# Disturbance in Some Fertility Biomarkers Induced and Changes in Testis Architecture by Chronic Exposure to Various Dosages of Each of Nonylphenol or Bisphenol A and Their Mix

**DOI:** 10.3390/life12101555

**Published:** 2022-10-07

**Authors:** Sahar J. Melebary, Mariam S. AlGhamdi, Manal E. A. Elhalwagy, Soha A. Alsolmy, Al Jawaher A. Bin Dohaish

**Affiliations:** 1Department of Biology, College of Science, University of Jeddah, Jeddah 21493, Saudi Arabia; 2Department of Biochemistry, College of Science, University of Jeddah, Jeddah 21493, Saudi Arabia

**Keywords:** Bisphenol A, Nonylphenol, oxidative stress markers, antioxidant markers, testis biomarkers, inflammatory markers

## Abstract

**Simple Summary:**

In this study, 70 male albino rats were allocated to the control group (GI) and were given 1 mL of ethanol; G, II, and G III were treated with 100 mg/Kg of each of BPA and NP, G IV and G V were treated with 25 mg/Kg of each of BPA and NP, G VI was given a mixture of a high dose of BPA and NP, and G VII was given a mixture of a low dose of BPA and NP. A significant elevation in the TNF Alpha, TNF Beta, and Caspase 3 serum was recorded individually and in the groups treated with high doses. In conclusion, exposure to BPA and NP strongly impacts antioxidants, immune-inflammatory mediators, and testicular tissue architecture. Furthermore, the data from this investigation support the idea that exposure to BPA and NP in daily life has multiple damages.

**Abstract:**

This investigation was conducted to demonstrate the potential impacts of different doses of Bisphenol A (BPA) or Nonylphenol (NP) and their mixtures on some biological activities in male albino rats. Seventy male albino rats were allocated to the control group (GI) and were given 1 mL of ethanol. G II and G III were given 100 mg/kg of each of BPA and NP, G IV and G V were given 25 mg/kg of each of BPA and NP, G VI was given a high dose of BPA and NP, and G VII was given a low dose of BPA and NP. All animals were treated orally for 60 days. Serum biomarkers of oxidative stress, antioxidants, immune-inflammatory mediators, and apoptotic markers were determined, as well as a histopathological examination of the testis at the end of the experimental period. The results obtained showed a pronounced increase in malondialdehyde (MDA), protein carbonyl (PC), and 4-hydroxynonenol (4-HNE), concomitant with a significant reduction in serum Superoxide dismutase (SOD), catalase enzyme (CAT), and total antioxidant capacity (TAC) in all treated groups. A significant elevation in TNF Alpha, TNF Beta, and Caspase 3 serum was recorded individually and in the groups treated with high doses. The disturbance is represented by histological damage in the testis in the germinal epithelium and a decrease in spermatozoa inside the lumen of seminiferous tubules. The effects on testis tissues were dose-dependent, pronounced in mixture doses, and remarkable in higher doses. In conclusion, exposure to BPA and NP strongly impacts antioxidants, immune-inflammatory mediators, and testis tissue architecture. Furthermore, the data from this investigation support the idea that exposure to BPA and NP in daily life has multiple damages.

## 1. Introduction

The global use of plastic is increasing, as it is an inert and cheap material. The quantity of mishandled plastic garbage in the ecosystem was determined, globally, to be 60–99 million tons in 2015 [1]. Plastic outputs include different materials depending on the usage, such as polypropylene, polyvinylchloride, polystyrene, and polyethylene [2]. There is an increasing worry about endocrine-disrupting chemicals spreading in the ecosystem.

Nonylphenol (NP) and bisphenol A (BPA) were found in both in vivo and in vitro trials to work as estrogenic materials. Synthetic xenoestrogens contain widely utilized industrial materials, such as octylphenol, nonylphenol, and bisphenol A (BPA), which are estrogenic chemicals capable of reacting with human estrogen receptors (ER) [3,4,5]. BPA is a building unit of polycarbonate plastics, a hard plastic applied to produce different human needs, including even baby products and water containers. Its terminal output contains coatings, adhesives, paints, and building compounds. BPA can be introduced to the human body via reusable dental sealants, baby bottles, the liquid of canned vegetables, and food packing materials [6]. 

Studies revealed that BPA could leach out of some products involving the plastic lining of cans utilized for food, tableware, polycarbonate baby bottles, and white dental filling sealants.

The adverse impacts of BPA are primarily due to its estrogenic effects. Their actions are mediated by endocrine signaling pathways, resulting in significant changes in cell functions even at low concentrations. These changes have deleterious effects on health, including prostate and breast cancer. It has been reported that BPA induces oxidative stress in different body organs [5].

Nonionic surfactants, or NPs, are a kind of nonionic surfactant frequently utilized in individual care applications such as cleansers, detergents, paints, cosmetics, intravaginal spermicides, hair colors, pesticides, and a variety of other synthetic items; it is also in polyvinyl chloride (PVC), which pollutes the water that passes via PVC pipes [7]. Nonylphenol contacts may interfere with pubertal development, and pubertal females appeared much more sensitive than males to being in contact with endocrine disruption chemicals (EDCs). Research revealed that some alkylphenols are estrogenic in birds, fish, and mammals. Precocious sexual growth may be induced by estrogenic materials found in the ecosystem. For NP-exposed prepubertal rats, there was an elevation in the vaginal opening, disruptions in estrous cyclicity, and alterations in the pituitary hormone [8].

Because of the lipophilic feature of NP, it precipitates in fat cells. Thus, it can be introduced to the food chain. Methods of contact with NP involve absorption, integument contact, eye contact, inhalation, and the oral route, and tropism organs for NP involve the eyes, integument, gut, respiratory tract, hepatic tissues, encephalon, thyroid, pancreas, renal tissues, bladder, and breast. In females, it can induce early puberty and ovary, breast, or endometrium cancer. In males, it can induce prostate, sperm, and seminiferous problems, damage in the gonads, the testis, the epididymal lumen, and the Sertoli cells, or cryptorchidism. It was recorded that 0.2–0.3 ng/mL of NP was noticed in the plasma samples obtained from healthy human volunteers [7].

The study was conducted to examine the potential impacts of different doses of Bisphenol A (BPA) or Nonylphenol (NP) and their mixtures on oxidative stress markers, antioxidant markers, anti-inflammatory markers, and apoptotic markers, as well as the effects on testicular biomarkers and changes in their tissue architecture in male albino rats.

## 2. Materials and Methods

### 2.1. Chemicals

1. Bisphenol A (BPA): White crystal—powder used for laboratory research purposes. It was obtained from TCI America.

Chemical material: 2,2-Bis(4-hydroxyphenyl) propane, Chemical formula: C_15_H_16_O, CAS RN: 80-05-7

The doses used in this study were chosen according to acceptable daily intake: a high dosage of 100 mg/kg b. w. and a low dosage of 25 mg/kg b. w. [9]. 

2. 4-Nonylphenol: Liquid, Clear, Colorless-Pale yellow, used for laboratory research. It was received from TCI America.

Chemical name: 4-Nonylphenol, Chemical formula: C_15_H_24_O, CAS RN: 84852-15-3

The doses used in this study were chosen according to acceptable daily intake: a high dosage of 100 mg/kg b. w. [10] and a low dosage of 25 mg/kg b. w. [11].

### 2.2. Experimental Design

As seen in Figure 1, 70 adult male Albino Wistar rats weighing 150–200 g were acquired from the Animal House at King Fahd Center for Medical Research at King Abdulaziz University and were confirmed by the Approved of Ethics Committee of the university. The animals were allocated into seven experimental groups, each group containing 10 rats, and the experimental period was 2 months. The groups were classified as: 

G I, (Control rats): Rats were given water with 0.02% ethanol and kept as the (+ve) control; G II, (BH): Animals were supplied orally with an elevated dosage (100 mg/Kg) of Bisphenol A; G III, (BL): Animals were supplied orally with a low dosage (25 mg/Kg) of Bisphenol A; G IV, (NH): Animals were supplied orally with an elevated dosage (100 mg/Kg) of 4 Nonylphenol; G V, (NL): Animals were supplied orally with a small dosage (25 mg/Kg) of 4 Nonylphenol; G VI, (MIX of BH & NH): Animals were supplied orally with a mix of elevated dosages (100 mg/Kg) of each of BPA and NP; and G VII, (MIX of BL & NL): Animals were supplied orally with small dosages (25 mg/Kg) of each of BPA and NP. The study was extended for 2 months.

### 2.3. Samples

At the end of the experiment, rats fasted overnight, and blood samples were collected from the retro-orbital plexus vein into nonheparinized tubes. Blood samples were centrifuged at 2000 rpm for 15 min to obtain serum. The serum was separated and kept in a deep freezer at −20 °C until the assays were carried out.

### 2.4. Blood Biochemical Markers

#### 2.4.1. Oxidative Stress Markers

The malondialdehyde MDA level was estimated following Yoshioka et al. [12], protein carbonyl (Pc) was determined following Cadenas et al. [13] and Wakeyama, et al. [14], and for 4-hydroxynonenal (4-HNE), the Enzyme-Linked Immunosorbent Assay (ELISA) Kit utilized the competitive enzyme immunoassay technique using a polyclonal anti-4-HNE antibody (Ab) and a 4-HNE-HRP conjugate.

#### 2.4.2. Antioxidant Biomarkers

Superoxide Dismutase (SOD) was estimated following Minami and following Minami and Yoshikawa [15], Catalase (CAT) was estimated following Aebi [16], and Total Antioxidant Capacity (TAC) was estimated following Koracevic et al. [17].

#### 2.4.3. Testis Biomarkers

Testosterone and luteinizing hormone (LH) were evaluated based on the Competitive-ELISA detection method.

#### 2.4.4. Inflammatory Apoptic and Markers

Nuclear Factor Alpha (NF α) The Picokine™ Rat Tnf Pre-Coated ELISA kit is a solid stage immunoassay particularly formed to estimate rat NF with a 96-well strip plate that is pre-coated with specific Ab for NF. 

Nuclear Factor Kappa B (NF-κB) The kit was obtained from My Bio Source, NC. company diagnostic and research reagents (Southern California, San Diego (USA)). The assay was carried out using the method of Perkins [18].

Caspase 3 (CASP3) used this Quantitative Sandwich ELISA applied for the estimation of the degree of CASP3 in undiluted original rat body fluids, secretions, tissue homogenates, or feces specimens. 

### 2.5. Histopathology

Animals were dissected for testis; the dissected organs were weighed, washed in normal saline, and fixed in 10% formalin solution for histopathological investigation. The processing method for light microscopic testing (hematoxylin and eosin staining) followed Carleton et al. [19].

### 2.6. Statistical Evaluation

The obtained data were statistically evaluated using SPSS statistical software package version 22. A two-way ANOVA tested all the data. The level of significance was set at LSD at *p* < 0.05 and Duncan’s from the control.

## 3. Results

### 3.1. Oxidative Stress Biomarkers in Serum

The data in Table 1 declared that the lipid peroxidation biomarker malondialdehyde (MDA) recorded a significant increase in the (NH) group versus (BL) at (*p* < 0.05). When compared to single material-supplied groups, primary-while-mixed groups with both high (BH and NH) and low (BL and NL) doses revealed a remarkable significant increase in MDA. The elevation percentage was expressed as 72.81% versus 22.81% from the control

The protein oxidation biomarker PC had the same MDA results as the MDA results, with significant elevation reported in all groups. A pronounced increase was reported in (BH), (NH), and their mixtures (BH and NH), with a percentage change from the control of 42.47%, 24.27%, and 100%, respectively. On the other hand, the percentage changes in the increase in low doses (BL & NL) and their mixture were 18.20%, 11.1%, and 43.59%, respectively.

On the other hand, the 4-hydroxynonenol (4-HNE) marker recorded a remarkable increase in high doses (BH and NH), and their mixtures were significant versus the control and other groups. When (BL) and (NL) were used, the amount of 4-HNE in the treated groups was elevated significantly compared to the control and high-dose-treated groups. 

### 3.2. Antioxidant Biomarkers in Serum

Table 2 illustrates the effect of supplying high and low dosages of each of BPA (BH) and (BL) and NP (NH) and (NL), as well as their mixture, on the serum antioxidant biomarkers. The data show a marked decrease in serum SOD throughout the experimental groups, with a significant difference between the control and all other groups (*p* < 0.05).

Moreover, the CAT enzyme levels revealed a significant decrease in all supplied groups compared to the control and all other groups. A mild increase (7.17%) from the control was reported in the (NL)-supplied group, but this elevation was statistically significant against the control (BH), (BL), and (NH). Regarding TAC, the results revealed a remarkable and significant decrease in sera via the treatment groups, which was significant against the control and among groups. It is worth mentioning that the noticeable decrease was noticeable in the mix (BH & NH) groups.

### 3.3. Serum Testis Biomarkers

The data presented in Table 3 show the treatment with large and small levels of each of BPA (BH & BL) and NP (NH & NL) and their mix of serum testosterone and luteinizing hormones (LH). Regarding the testosterone hormone, the demonstrated results revealed a pronounced significant reduction in testosterone in the high-dose-treated groups (BH, NH, and the mix of BH and NH); the significance versus the control and between groups was at *p* < 0.05. The low-dose-treated groups (BL, NL, and the Mix of BL & NL) recorded less of a reduction in hormone levels compared to the high-dose groups at (*p* < 0.05).

Regarding the luteinizing hormone (LH), the illustrated results showed a dose-dependent elevation in hormone levels in all the treatment groups, which was significant against the control and the NH and NL groups in high- and low-treated mixtures.

### 3.4. Testis Histological Result

Despite the slight decrease in the BL and NL groups, a noticeable and significant elevation was recorded in the BL and NL groups. The serum caspase biomarker results revealed that the high-dosage mixture group (Mix H) induced a significant increase in the caspase level by 46.91% compared to the control, and this was significant compared to all other supplied groups (*p ≤* 0.05). A marked decrease in serum caspase in the BL-, NL-, and Mix L-treated groups was detected with (−76%, −54%, −79.01%, and −35.580%), respectively, which was significant versus the control and high-dose-treated groups (Table 4).

The histological investigation of the rats’ testes in the control animals showed that the testicular histoarchitecture follows the standard structure. In addition, the normal cellular architecture of seminiferous tubules and the abundant spermatids in the lumen were observed in the control testis (Figure 2a). On the other hand, different levels of severity and damage were observed in the testes of the adult rat’s gavage to low doses of 25 mg/kg (b. w.) of Bisphenol and Nonylphenol separately and in their mixture. The interstitial tissue in the rat’s gavage with low doses of BPA appeared less dense, with a widening in the interstitial space, lymphocytic infiltration, and Leydig cells appearing in groups, some of which were degenerated and necrotic. The seminiferous tubules appeared in different shapes, including round, oval, and irregular, with a change in the thickness of the basement membrane (BM) from one part to another. There was an increase in the pigmentation of myoid cells (MC) (Figure 2b).

In Figure 2c, the seminiferous tubules in the sections of the tissue gavage with a low dose of NP appeared oval-shaped, with a widening of the lumen, and they were irregular in shape, with a varying thickness from one part to another. There was a slight increase in the size of the interstitial connective tissue and the number of Leydig cells that were found in groups around the congested blood vessels.

Severe disruptions in the histoarchitecture of the seminiferous tubules were observed in the testes of the rats’ gavage with a low mixture dose of BPA and NP. Most germinal cells appeared deformed, lysed, sloughed, and separated from the tubular epithelium. Notably, dissociation between the sperm cells decreased spermatogenesis, where spermatogonia appeared in atrophic and necrotic nuclei. Leydig cells appeared to be dark with atrophic nuclei and vacuoles (Figure 2d).

However, in the histological sections of the testes of the rat’s gavage with a BPA dose of 100 mg/kg (b. w.), in Figure 2e, a severe increase in tissue damage was represented by a severe widening between the seminiferous tubules, with a decrease in the density of the interstitial tissue and dilation and congestion of the blood vessels. Some Leydig cells degenerated and lost their standard shape. An increase in the number of primary spermatocytes in some seminiferous tubules and a decrease in the number of secondary spermatocytes and sperm indicate the termination of spermatogenesis, which led to the widening of the lumen of the seminiferous tubules (L) and less cellular association. The nucleus of spermatogonia also appeared atrophic and separated from the germinal layers. The Sertoli cells also lost their order and vertical shape from the basement membrane.

Figure 2f shows that some of the seminiferous tubules in the rat’s gavage with an elevated dosage of 100 mg/kg (b. w.) of NP were degenerated and irregular in shape, with varying thicknesses of cell walls in some parts. Some others revealed an increase in the number of primary spermatocytes and a decrease in others, with a widening in the interstitial space and degeneration and hemorrhage in some places. Additionally, the congestion of blood vessels and the nuclei of Leydig cells appeared dark and atrophic.

In addition to the previous damage, an acute increase in the severity of the tissues in the damage in Figure 2g was noticed in the testes of the rat’s gavage with a high dosage of 100 mg/kg (b. w.) of BPA and NP mixtures. Spermatogonia appeared to be few in number and deeply stained, and many had degenerated cells and pyknotic nuclei. Many spermatogonia have a deep rounded nucleus surrounded by a clear halo area of cytoplasm (apoptotic cells). Additionally, there is a clear decrease in the number of primary spermatocytes, which are dying off, and there are only a few mature spermatids with bodies left over.

## 4. Discussion 

Oxidative stress (OS) is a case of disturbance among oxidants and antioxidants in favor of oxidants due to the induction of free radicals. Free radicals act as a critical action in cellular destruction arising from using poisonous chemicals, which induce cell necrosis [20]. OS also happens due to the elevation of ROS, which can induce significant cell destruction via reacting with many molecules involving proteins, fats, and DNA, controlling lipid peroxidation. Thus, it has a critical action in the pathogenesis of many human health disorders [21]. Different ecological pollutants can cause OS by stimulating the liberation of ROS, such as hydrogen peroxide (H_2_O_2_) and superoxide anion [22,23].

The formation of MDA is a significant marker of OS, which, with a decrease in the cell’s anti-oxidative activity, can destroy the cell membrane. These are the reported findings from rats supplied with various levels of Nonylphenol [24]. Similarly, in in vivo studies, a range of BPA levels have shown a significant reduction in the TAC in several cells and organs involving hepatic tissue, testicles, and the pancreas [25,26], and a lowering of SOD, CAT levels, and GPx was also recorded in the encephalon, epididymal sperm, hepatic tissues, pancreas, renal tissues, testes, and germ cells. The lowering of antioxidant levels is related to the stimulation of ROS by BPA over a different dosage, as noticed for ROS stimulation. There are changes in the enzymatic and non-enzymatic antioxidant activities in every cell, tissue, and organ [27]. When antioxidant defenses decline, body cells become more susceptible to dysfunction and disease conditions. 

The MDA levels could correlate with the cell’s rate and intensity of lipid peroxidation [28]. Different reports showed associations between urinary bisphenol levels and markers of OS, isoprostane, and MDA [29]. Additionally, Dutta and Paul [30] reported a significant increase in protein carbonyl in rats treated with BPA, and the elevation was dose- and time-dependent. BPA contact concentrations were related to 8-OHdG, which indicated that ecological BPA contact could trigger or escalate oxidative DNA destruction in the tissues [31,32]. 

It is exciting that BPA is considered a pro-oxidant and an antioxidant, as BPA’s structure has antioxidants in manufactory impacts, while it appears to have pro-oxidant action via its metabolites or the estrogen receptor. Free radical production via metabolic redox cycling among the quinone and hydroquinone forms of BPA may induce oxidative damage [33]. 

In the meantime, supplying NP at a dosage of 50 µg/kg bw/day for a month revealed an elevation in OS markers in the sera of male rats [34]. As the two phenolic materials stimulate similar functions, the marked increase in all OS markers in the mix groups can indicate the synergistic impact of the two materials that were more significant in the large dosage mix. In parallel to the mechanism mentioned above, a decrease in the assessed serum antioxidant markers SOD, CAT, and TAC was significantly reported in the current investigation in mix-supplied animals. These data concur with an experiment recording that the supplementation of NP at 15,150 and 1500 lg/kg b. w. per day for a month and a half revealed a dose-dependent elevation in the concentration of H_2_O_2_ and a lowering of antioxidant enzymes in the hepatic tissue of supplied rats [11].

Regarding the effects of BPA and NP and their mixtures on the level of serum testosterone and luteinizing hormone in treated rats, the obtained results declared a pronounced significant reduction in testosterone in the high-dose-treated groups (BH, NH, and Mix of BH & NH), while the low-dose-treated groups (BL, NL, and Mix of BL & NL) recorded a less pronounced reduction in hormone levels. However, an unexpected dose-dependent elevation in LH hormone levels throughout the treated groups was recorded in the present study. The histopathological findings of the testis elucidate the deleterious effects of high doses of phenolic compounds and their mixture on the testis, where dilation of the blood vessels with thick walls appeared, and a reduction in the number of Leydig cells was recorded. 

BPA acts as an estrogen against androgen antagonist activity [35]. BPA-supplying S/C for 15 days induced a marked reduction in testosterone levels and increased luteinizing hormone levels in adult rat testicles and sera [36]. BPA lowers testosterone and the size of seminiferous tubules, revealing the disintegration of germinal epithelial cells and spermatogenesis [37]. This finding runs parallel with our results. Likewise, rats receiving 250 mg/kg of NP showed a reduction in testosterone concentration, but their LH and FSH were elevated [38]. 

Omran et al. [39] declared that the elevation of the level of testosterone and the damage to the male reproductive system are caused by the actions of xenoestrogen BPA on the formation of gonadal steroidogenesis. Similar results were reported by Qui et al. [40] and Kazemi et al. [41]. NP and BPA have an inactivation impact on the P450 cytochrome, a critical enzyme in the production of testosterone in Leydig cells. In contrast, two-month-old male rats, supplied with 250 mg/kg of NP for 12 days, revealed an elevation of testosterone and FSH levels; however, LH was not affected [42]. 

Chronic contact with NP decreases the size of the testis, decreases the testosterone level, lowers the sperm count in the epididymis, lowers the size of the seminiferous tubule and the lumen, leads to epithelial thickness, cryptorchidism, elevated Sertoli cell apoptosis, and the hypertrophy of Sertoli cells, depresses the action of antioxidant enzymes in the epididymal sperm, disturbs the testicular structure, and induces testis tumors [43,44]. These earlier findings run parallel with our results and confirm histopathological studies. As stated above, expectedly, both BPA and NP disturbed the endocrine system of the supplied animals [45]. Rats subjected to a small dosage of a mix between BPA and NP exhibited deformation and lysis of the germinal cells, with separation of the tubular epithelia and the occurrence of atrophic and necrotic spermatogonia and less cellular association, where some cells appeared apoptotic, with an oval halo of cytoplasm and deeply stained nuclei [41]. 

The current investigation results concur with those of Su et al. [46], as in vivo trials showed that both excess autophagy and apoptosis are associated with testicular destruction and the intoxication of prepubertal rats post-contact with small doses of BPA and NP. Considering the results of the pro-inflammatory markers TNF-alpha and TNF-KB depicted in the present study, a pronounced effect was recorded in the high-dose-treated groups, either individually or combined. Furthermore, the caspase3 apoptosis marker studied in the current study showed a significant increase in the high-dose-treated groups in individual BH or NH and in combined BH & NH. 

According to Jung et al. [47], oxidant generation increased NF-B activation, which increased TNF levels and caused tissue injury. NF-κB leads to nuclear translocation, which regulates specific cytokine genes such as TNF-alpha [48]. BPA can modify different transcriptional factors, such as Sp1 (specificity protein 1), that affect the combining site of NF-κB and hence change the transcriptional capacity of NF-κB [49]. 

NP improves apoptosis by sera deprivation in PC1_2_ cells via caspase 3 [50]. Furthermore, Liu et al. [51] reported that elevated apoptosis with the increased expression of caspase 3 was noticed in the testes of mice pups subjected to BPA through pregnancy and lactation. The various data may be clarified by the probability that NP causes apoptosis in various cells via different techniques.

The lipophilic property of Nonylphenol may reveal severe health complications. Experiments showed a marked decrease in the viability of thymocytes, adipocytes, tumor cells, embryonic stem cells, and Sertoli cells because of enhanced programmed cell apoptosis [24] and also revealed the stimulation of the initiator caspase 8 [52]. The resulting complexes bind and activate pro-caspases-8 and 10 [53]. Additionally, it directly stimulates effector caspases, such as caspase 3. These findings explain the previous descriptive results, as the elevation of the caspase 3 level is concurrent with the elevation in inflammatory marker cytokines, especially in high-dose-treated groups.

Saied and Darwish [54] and Yousaf et al. [55] mentioned that changes in the testicular characteristics are an indicator of environmental pollution by BPA. Nakamura et al. [56] reported that male Wistar rats’ subcutaneous administration of BPA for a month and a half reduced the number of Leydig cells and the testosterone levels. This variation may be attributed to differences in animal species. Gurmeet et al. [57] mentioned that BPA exposure impacts the development and function of the reproductive organs at puberty and throughout adulthood. 

At high concentrations of BPA (100 mg/kg (b. w.)), the data of the current investigation are in agreement with Munir et al. [58]. They declared that, under the impact of BPA, the arrangement of spermatogenesis was disrupted. No mature sperms in the lumen were noted; however, residual bodies were found. Hutan [59] reported that spermatids decreased due to a disruption in spermatogenesis in rats treated with high concentrations of BPA.

These results are in agreement with the current study and the studies of Hassan et al. [60], Xi et al. [61], and Nakamura et al. [56]. In a previous study, the intercellular bridges between germ and Sertoli cells were loosened in animals treated with BPA [57]. The damage in the testis may be due to BPA, characterized by xenoestrogen properties and inhibiting testicular glands’ growth [39]. BPA selectively targets the male reproductive organ, directly affecting the testicular function [62]. 

Karumari and Balasubramanian [63] noticed a deformation in spermatid formation in the testis because of spermatogenic inhibition in different seminiferous tubules in rats treated with BPA. Furthermore, cellular changes such as the widening of the seminiferous tubules and edema were revealed. Kazemi et al. [41] confirmed that a reduction in the testosterone level revealed a lowering of the count of spermatocytes, spermatids, and spermatogonia when male rats were subjected to ecological BPA. BPA reduces testosterone levels and the diameter of seminiferous tubules and destroys germinal epithelial cells and spermatogenesis [37]. Additionally, Lee and Rhee [64] mentioned that BPA has an antiandrogenic effect by blocking the action of dihydrotestosterone.

The results of Kazemi et al. [41] agree with the present study, where degenerative changes were found in the Sertoli cell nucleus and cytoplasm. These Sertoli cell deformations might affect spermatogenic cells’ normal maturation stages and nutritional intake. This study’s results showed the elevation of the testosterone concentration and damage to the male reproductive system. 

NP works on inhibiting the action of endocrine disruptors due to its ability to cause harmful reproductive effects on mammals by causing cell death of the reproductive glands [43]. Trudeau et al. [65] mentioned that this could be attributed to estrogenic chemicals inhibiting testicular growth due to their direct effect on the testis by inhibiting androgen synthesis (which is required for spermatogenesis). Tabassum et al. [66] mentioned that NP has a weak connective power for estrogen receptors. In a study by Hu et al. [67] on male rats, in vivo or in vitro, it was revealed that NP disrupts the structure and functions of Sertoli cells and reproduction hormones in serum at low doses. Supplementation with NP elevates the apoptosis of Sertoli and germinal cells, lowering the formation of sperm [38].

Duan et al. [68] mentioned that the exposure to NP before maturity in mice disrupts the reproductive system during maturity. Various studies [69,70,71] confirmed that spermatogenesis, sperm function, and morphology are affected when treated with NP. Dalgaard et al. [72] and Kinnberg and Toft [73] explained that a high exposure to NP causes a blockage of spermatogenesis, causing degenerative changes in Leydig cells and the deterioration of Sertoli cells due to the disruption of cyst formation. Jobling et al. [74], Miles-Richardson et al. [75], and Panter et al. [76] reported the effects of exposure to natural or environmental estrogen on male fish during sexual maturation.

These well-documented studies discuss the delay in spermatogenesis and the consequent effects on gonadal development due to its presence in known concentrations in effluents [77] and fish of polluted rivers [78]. NP inhibits lipid metabolism, which reduces testosterone production, causing adverse effects for spermatogenesis and harmful effects for Leydig cells.

Interestingly, NP leads to testicular apoptosis in addition to oxidative stress [79,80]. Duan et al. [68] reported that the exposure of rats to 60 mg/kg of NP caused severe destruction of the seminiferous tubule and spermatogenesis derangement, and a large dose of NP caused a disorder in the average balance between cell proliferation and apoptosis. Additionally, oxidative stress contributed to the caspase activation that works on testicular apoptosis due to NP. Peng et al. [68] explained that this could be attributed to the vital role of caspase signals in activating apoptotic signals. 

Ahn et al. [81] mentioned that mitochondrial and death receptor pathways refer to the molecular mechanisms attributed to apoptosis due to caspase activation. The exposure of the rats to NP leads to understanding its influence on male fertility through studying the results of harmful effects of hormone deficiency, oxidative testicular damage, the inhibition of cell proliferation, the comptonization of sperm and testicular function, and inducing testicular germ cell apoptosis. This investigation’s findings concur with those of Su et al. [46], as in vivo trials revealed that both excess autophagy and apoptosis are correlated with testicular destruction and the intoxication of prepubertal rats post-contact with trim levels of BPA and NP.

The present study showed that the exposure to 100 mg/kg caused a clear decrease in the number and degeneration of germinal cells, with numerous changes represented in spermatogonia apoptosis and necrosis, with a decrease in the number of spermatids, the presence of residual bodies, the widening of the interstitial space, and a decrease in the number of Leydig cells.

The serum testosterone level was lowered in the groups supplied with BPA and NP, followed by Kazemi et al. [41]. Leonard et al. [82] mentioned that cellular apoptosis is an early indicator of xenobiotic stress and can offer information related to the quality of the living organism. The opposite impact of low and high doses is recorded, as many of the impacts that occur in response to low doses of hormones do not occur at much higher doses. Vom Saal et al. [83] mentioned that the dose utilized in both in vitro and in vivo trials must be considered in evaluating the findings’ physiological relevance.

## 5. Conclusions

The rat’s exposure to BPA or NP has substantial adverse impacts on antioxidants, immune-inflammatory mediators, and testis tissue architecture. The mixture between BPA and NP revealed a synergistic negative impact in all the examined parameters. The data of this investigation support the idea that exposure to BPA and NP in daily life has multiple damages and should be investigated in future studies with the suggestion of possible procedures that could be applied to reduce or overcome such adverse effects.

## Figures and Tables

**Figure 1 life-12-01555-f001:**
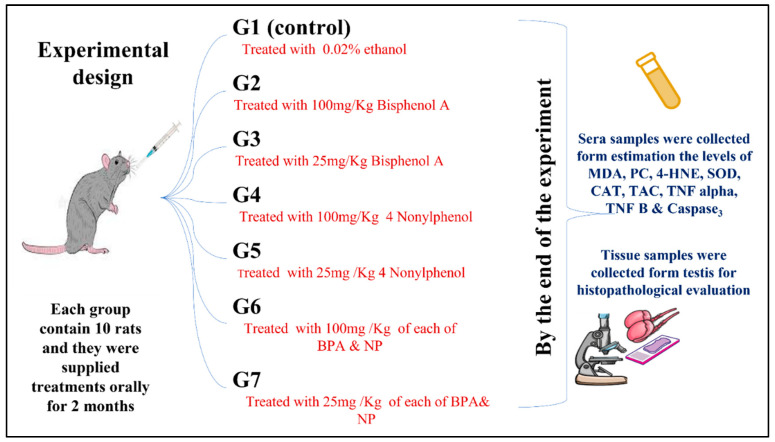
Experimental design.

**Figure 2 life-12-01555-f002:**
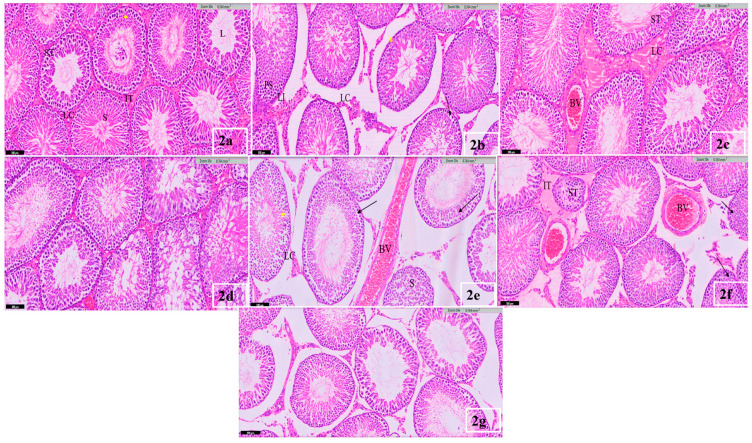
(**a**) Light micrograph of the control showing: seminiferous tubules (ST) with germ cells in various stages of spermatogenesis (*); healthy sperms (S) are seen in and around the tubular lumen (L); interstitial tissue (IT) containing Leydig cells (LC). H & E (×100 μm). (**b**) Microscopical appearance of animals treated with 25 mg/Kg (b. w.) of Bisphenol A, revealing: abnormal histoarchitecture with less dense interstitial tissue containing numerous Leydig cells (LC); lymphocytic infiltration (LI); basement membrane is thick in some parts (*). Notice the intra-cellular vacuoles of different sizes between the germ cells (arrow); an increase in the number of primary spermatocytes in some seminiferous tubules (PS). H & E (×100 μm). (**c**) Microscopical appearance of animals treated with 25 mg/Kg (b. w.) of Nonylphenol, revealing: congestion of blood vessels (BV) in the interstitial tissues; the presence of Leydig cells in groups (LC); seminiferous tubules are oval-shaped (ST); the basement membrane is irregular and varies in thickness from one part to another. Notice the germinal cells at various developmental stages. H & E (×100 μm). (**d**) Microscopical appearance of animals treated with 25 mg/Kg (b. w.) of the mix of Bisphenol A and Nonylphenol, revealing: disruption in seminiferous tubules’ histoarchitecture. Notice the deformation and lysis of germ cells; the separation of the tubular epithelium. Spermatogonia appeared atrophic and necrotic. H & E (×100 μm). (**e**) Microscopical appearance of animals treated with 100 mg/Kg of Bisphenol A, revealing: widening of the interstitial space with a less dense interstitial tissue; hyperplasia and congestion of the blood vessel (BV); Leydig cells in groups (LC); disruption and sloughing of the germ cells (arrow). Notice the increase in the number of primary spermatocytes in some tubules with a decrease in the number of spermatids (S); loss of spermatocytes in other tubules. Spermatogonia appeared atrophic and separated from the germinal layer (*). H & E (×100 μm). (**f**) Microscopical appearance of animals treated with 100 mg/Kg (b. w.) of Nonylphenol, revealing: various degrees of damage and degeneration of some seminiferous tubules (ST). Notice the increase in the quantity of primary spermatocytes (arrow) in some seminiferous tubules; widening of the interstitial space; lysis and edema of the interstitial tissue (IT); congestion of blood vessels (BV). H & E (×100 μm). (**g**) Microscopical appearance of animals treated with 100 mg/Kg (b. w.) of mixed levels of Bisphenol A & Nonylphenol, revealing: disruption in the stratified germinal cells; a marked decrease in the number and degeneration of germinal cells. Notice the necrosis and apoptosis of spermatogonia; irregularities in some tubular walls; the interstitial cells of Leydig are few in number and degenerated; widening of interstitial space. H & E (×100 μm).

**Table 1 life-12-01555-t001:** Impact of supplying various levels of BPA and/or NP on oxidative stress markers in the sera of albino rats.

	Parameters	MDA nmol/mL	PC μmol/mg	4-HNE ng/mL
Groups				
Control		1.14 ± 0.06	8.90 ± 0.29	194.40 ± 5.71
Bisphenol A high dose (BH)		1.16 ± 0.04	12.68 ± 0.49 ^a^	218.00 ± 8.69 ^a^
Bisphenol A low dose (BL)		1.01 ± 0.02	10.52 ± 0.21 ^b^	145.20 ± 7.99 ^ab^
Nonylphenol high dose (NH)		1.22 ± 0.03 ^c^	11.06 ± 0.83 ^a^	206.60 ± 6.54 ^c^
Nonylphenol low dose (NL)		1.11 ± 0.05	9.88 ± 0.38 ^b^	183.60 ± 5.82 ^bcd^
Mix (BH & NH)		1.97 ± 0.05 ^abcde^	17.80 ± 1.33 ^abcde^	306.60 ± 7.59 ^abcde^
Mix (BL & NL)		1.40 ± 0.06 ^abcdef^	12.78 ± 0.65 ^acef^	237.60 ± 10.08 ^acdef^

Malondialdehyde (MDA); protein carbonyl (PC); 4- Hydroxynonenal (4-HNE). All data were expressed as the mean ± SE of all groups (10 rats each). ^a^: significance against control at *p* < 0.05; ^b^: significance against BH G at *p* < 0.05; ^c^: significance against BL G at *p* < 0.05; ^d^: significance against NH G at *p* < 0.05; ^e^: significance against NL G at *p* < 0.05; ^f^: significance against Mix (BH & NH) G at *p* < 0.05.

**Table 2 life-12-01555-t002:** Impact of supplying various levels of BPA and/or NP on antioxidant markers in the sera of albino rats.

	Parameters	SOD u/mL	CAT Mu/L	TAC μmol/mg
Groups				
Control		303.60 ± 4.75	92.00 ± 3.00	38.00 ± 2.24
Bisphenol A high dose (BH)		219.80 ± 7.41 ^a^	83.70 ± 4.62 ^a^	30.16 ± 4.17 ^a^
Bisphenol A low dose (BL)		274.00 ± 14.15 ^b^	88.66 ± 1.85 ^ab^	31.26 ± 1.99 ^ab^
Nonylphenol high dose (NH)		226.80 ± 6.92 ^abc^	88.80 ± 3.71 ^c^	26.16 ± 1.58
Nonylphenol low dose (NL)		207.80 ± 7.48 ^abc^	98.60 ± 1.63 ^abd^	32.70 ± 1.67
Mix (BH & NH)		220.60 ± 10.99 ^abcde^	48.80 ± 3.43 ^abcde^	21.40 ± 1.39 ^abcde^
Mix (BL & NL)		274.60 ± 17.53 ^bdef^	82.60 ± 5.27 ^bcef^	30.32 ± 1.91 ^abcde^

Superoxide dismutase (SOD); Catalase enzyme (CAT); Total antioxidant capacity (TAC). All data were expressed as the mean ± SE of all groups (10 rats each). ^a^: significance against control at *p* < 0.05; ^b^: significance against BH group at *p* < 0.05; ^c^: significance against BL group at *p* < 0.05; ^d^: significance against NH group at *p* < 0.05; ^e^: significance against NL group at *p* < 0.05; ^f^: significance against Mix (BH & NH) group at *p* < 0.05.

**Table 3 life-12-01555-t003:** Impact of supplying various levels of BPA and/or NP on testis biomarkers in the sera of albino rats.

	Parameters	Testosterone ng/mL	LH mIU/mL
Groups			
Control		181.62 ± 10.09	9.69 ± 0.23
Bisphenol A high dose (BH)		137.60 ± 5.69 ^a^	11.38 ± 0.57
Bisphenol A low dose (BL)		201.00 ± 7.09 ^b^	11.24 ± 0.97
Nonylphenol high dose (NH)		118.20 ± 4.02 ^ac^	13.26 ± 0.36 ^abc^
Nonylphenol low dose (NL)		179.60 ± 12.78 ^bd^	11.88 ± 0.39 ^a^
Mix (BH & NH)		100.20 ± 3.95 ^abce^	17.76 ± 1.05 ^abcde^
Mix (BL & NL)		160.80 ± 10.06 ^cdf^	13.60 ± 0.29 ^abcf^

Luteinizing Hormone (LH). All data were expressed as the mean ± SE of all groups (10 rats each). ^a^: significance against control at *p* < 0.05; ^b^: significance against BH group at *p* < 0.05; ^c^: significance against BL group at *p* < 0.05; ^d^: significance against NH group at *p* < 0.05; ^e^: significance against NL group at *p* < 0.05; ^f^: significance against Mix (BH & NH) group at *p* < 0.05.

**Table 4 life-12-01555-t004:** Impact of supplying various levels of BPA and/or NP on inflammatory markers in the sera of albino rats.

	Parameters	NF α pg/mL	NF-κB pg/mL	Caspase ng/mL
Groups				
Control		23.34 ± 2.07	48.08 ± 1.29	0.81 ± 0.06
Bisphenol A high dose (BH)		31.80 ± 5.75	56.21 ± 1.19 ^a^	0.84 ± 0.11
Bisphenol A low dose (BL)		19.66 ± 1.12 ^b^	40.72 ± 0.64 ^ab^	0.19 ± 0.03 ^ab^
Nonylphenol high dose (NH)		24.48 ± 2.20	55.06 ± 1.96 ^ac^	0.89 ± 0.07 ^c^
Nonylphenol low dose (NL)		19.68 ± 0.81 ^b^	44.90 ± 2.50 ^bd^	0.17 ± 0.04 ^abd^
Mix (BH & NH)		75.20 ± 3.11 ^abcde^	76.10 ± 3.63 ^abcde^	1.19 ± 0.10 ^abcde^
Mix (BL & NL)		45.00 ± 3.03 ^abcdef^	54.60 ± 2.04 ^acef^	0.52 ± 0.03 ^abcdef^

Nuclear Factor Alpha (NF- α); Nuclear Factor Kappa B (NF- κB); Caspase 3 (CASP3). All data were expressed as the mean ± SE of all groups (10 rats each). ^a^: significance against control at *p* < 0.05; ^b^: significance against BH group at *p* < 0.05; ^c^: significance against BL group at *p* < 0.05; ^d^: significance against NH group at *p* < 0.05; ^e^: significance against NL group at *p* < 0.05; ^f^: significance against Mix (BH & NH) group at *p* < 0.05.

## Data Availability

The data presented in this study are available on request from the corresponding authors.

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
