# Peer review of "Disturbance in Some Fertility Biomarkers Induced and Changes in Testis Architecture by Chronic Exposure to Various Dosages of Each of Nonylphenol or Bisphenol A and Their Mix"

_life, 2022, doi:10.3390/life12101555_

Round 1

Reviewer 1 Report

Thank you for asking me to provide a review of this article, which has a subject of high interest nowadays, as it is known that the contact with such substances like Bisphenol A (BPA), Nonylphenol (NP), their mixtures and also the Oxidative Stress (OS) may have an extremely agressive impact on the human body, as well as producing severe damage in the endocrin system. The analysis was conducted to examine the potential impacts of different doses of BPA, NP, or their mixtures on oxidative stress markers, antioxidant markers, anti-inflammatory and appoptotic markers, as well as the effects on the testis biomarkers and changes in its tissue architecture, on male albino rats. 

               Being an observational analysis, the study followed a group of 70 albino rats, divided in 7 groups, 6 of each treated with high or low doses of BPA, NP, or the mixture of these 2 components and the Control Group, which was given 0,02% ethanol. The study was conducted for a period of 2 months, which is quite low, so maybe it should have been better to be a larger period of time in order to gain more accurate information. 

              Regarding the structure and accuracy of the phrases, unfortunatelly the manuscript lacks in well structured information, and the phrases are not so well designed.

  The manuscript is original and well defined and  the results provide an advance in current knowledge. The results are being interpreted appropriately and are significant, but the writting techniques and the structure of the phrases are not well designed, and therefore, it is very difficult to read the article and to understand the meaning of the study and even the meaning of the phrases.  So unfortunatelly, the article is  not written in an appropriate way. 

             The data are robust enough, but it is very difficult to follow the conclusions due to the writting techniques. 

             Surely the paper will attract a wide readership, but only with the corrections being made. 

            The English language is not appropriate and has many writting mistakes, and so, it is imperative to be corrected so that the article could be well designed.

            I have some things to add in the lines below, but unfortunatelly, from my point of view, the article must be rewritten more carefully, especially regarding the English language: 

Line 13: 70, not „seventy”

Line 14: 1 ml of ethanol, not „1 ml ethanol”

Line 14: „.” after „ethanol”

Line 14: without „,” before „and GIII”

Line 14: were treated, not „treated”

Line 14: 100mg/kg, not „(100mg/kg)

Line 14: were treated, not „treated”

Line 15: 25mg/kg, not „(25mg/kg)”

Line 15: GVI was given a mixture of a high dose of BPA and NP, not „was given mixture (BPA and NP) of high dose”

Line 16: BPA and NP, not „(BPA and NP)”

Line 16: A significant elevation in the TNF Alpha, TNF Beta and Caspase 3 serum was recorded individually and also in the groups treated with the mixture of high doses, not „Significant elevation in serum TNF Alpha, TNF Beta and Caspase 3 was recorded individually and in a mixture of high doses treated groups”

Line 19: Furthermore, not „Further”

Line 19: supports that the exposure, not „support that exposure”

Line 24: 70, not „seventy”

Line 24: 1 ml of ethanol, not „1 ml ethanol”

Line 24: without „,” before „and GIII”

Line 25: were treated, not „treated”

Line 26: 25mg/kg, not „(25mg/kg)”

Line 26: GVI was given a mixture of a high dose of BPA and NP, not „was given mixture (BPA and NP) of high dose”

Line 27: BPA and NP, not „(BPA and NP)”

Line 29: „,” after „determined”

Line 30: The results obtained, not „Obtained results”

Line 33: A significant elevation in the TNF Alpha, TNF Beta and Caspase 3 serum was recorded individually and also in the groups treated with the mixture of high doses, not „Significant elevation in serum TNF Alpha, TNF Beta and Caspase 3 was recorded individually and in a mixture of high doses treated groups”

Line 34: The disturbance is represented by the histological, not „A disturbance represents the histological”

Line 36: The effects, not „Effects”

Line 36: „,” after „mixture doses”

Line 37: the exposure, not „exposure”

Line 38: Furthermore, not „Further”

Line 39: supports that the exposure, not „support that exposure”

Line 50: were found both in vivo, not „were found in both in vivo”

Line 56: involving even baby, not „involving almost baby”

Line 57: BPA can be introduced, not „BPA could be introduced”

Line 72: to be in contact, not „to contact”

Line 81: in females, it can induce early puberty, ovary, breast or endometrium cancer, not „in females, it can induce (early puberty in females, ovary, breast cancer, and endometrium)”

Line 82: in males, it can induce prostate, sperm, seminiferous problems, damage in the gonads, in the testis, epididymous lumen and in the Sertoli cells, or cryptorchidism, not „in males, it can induce (prostate, sperm, seminiferous problems, testis, epididymous lumen and epitelial, cryptorchidism, and Sertoli cells) damage in gonads”

Line 84: samples, not „sample”

Line 88: the effects, not „effects”

Line 108: 70, not „seventy”

Line 108: without „that rats”

Line 109: „King”, not „king”

Line 110: of the university, not „university”

Line 111: each group containing, not „each group contains”

Line 113: were given water, not „were taken water”

Line 114: were supplied, not „supplied”

Line 115: were supplied, not „supplied”

Line 116: were supplied, not „supplied”

Line 117: were supplied, not „supplied”

Line 119: were supplied, not „supplied”

Line 124: of the experiment, not „of experiment”

Line 125: retro-orbital, not „retro orbital”

Line 126: 2000G, not „2000g”

Line 126: the serums were, not „sera were”

Line 129: without „()”

Line 132: utilized, not „utilize”

Line 146: used this Quantitative..., not „this Quantitative...”

Line 146: only 1 space between „this” and „Quantitative”

Line 151: „,” after „saline”

Line 153: followed, not „follow”

Line 164: the percentage, not „%”

Line 166: had the same, not „has the same”

Line 184: „,” after „mixture”

Line 184: regarding, not „as regards”

Line 191: „,” after „groups”

Line 191: significant, not „Significant”

Line 201: that the treatment, not „that treatment”

Serum Testis Biomarkers

Lines 201-203: The meaning of the paragraph is not clear. Maybe it should have been written: „shows the treatment”, instead of „declare that treatment”

Line 203: Regarding the testosterone hormone, not „Testosterone hormone”

Line 222: high dose, not „high does’”

Line 222: induced, not „induce”

Line 236: „,” after „animals”

Line 243: were degenerated, not „are degenerated”

Line 246: There was an increase, not „An increase”

Line 267: and vertical shape, not „and vertical shape”

Line 268: shows that, not „show”

Line 268: without „,” before „some”

Line 331: which induces, not „which induce”

Line 337: marker, not „mark”

Line 338: „,” before „can destroy”

Line 340: have shown, not „was shown”

Line 345: without „look”

Line 345: activities in every cell, tissue..., not „activities look to be very cell, tissue...”

Line 349: in people, not „in peoples”

Line 353: elevate, not „elevation”

Line 355: „,” after „antioxidant”

Line 372: „,” before „while”, instead of „.” before „while”

Line 377: „,” after „appeared”

Line 398: the endocrin system, not „the endocrin”

Line 418: reported that, not „reported”

Line 425: and also revealed, not „.” after „[23]”

Line 457: by blocking, not „blocking ”

Line 463: the action of edocrin, not „the work of edocrin”

Line 464: by causing, not „causing”

Line 472: that the exposure, not „that exposure”

Line 472: „,” after „mice”

Line 486: that the exposure, not „that exposure”

Line 488: „.” after „apoptosis”

Line 488: Also, not „also”

Lines 493-494: The exposure of the rats, not „Exposure of rats”

Line 501: that the exposure, not „that exposure”

Line 519: that the exposure, not „that exposure”

Author Response

Reviewer 1

Comments and Suggestions for Authors

Thank you for asking me to provide a review of this article, which has a subject of high interest nowadays, as it is known that the contact with such substances like Bisphenol A (BPA), Nonylphenol (NP), their mixtures and also the Oxidative Stress (OS) may have an extremely agressive impact on the human body, as well as producing severe damage in the endocrin system. The analysis was conducted to examine the potential impacts of different doses of BPA, NP, or their mixtures on oxidative stress markers, antioxidant markers, anti-inflammatory and appoptotic markers, as well as the effects on the testis biomarkers and changes in its tissue architecture, on male albino rats. 

Being an observational analysis, the study followed a group of 70 albino rats, divided in 7 groups, 6 of each treated with high or low doses of BPA, NP, or the mixture of these 2 components and the Control Group, which was given 0,02% ethanol. The study was conducted for a period of 2 months, which is quite low, so maybe it should have been better to be a larger period of time in order to gain more accurate information. 

Response: thanks for your valuable comment regarding time, it will be considered in further work

Regarding the structure and accuracy of the phrases, unfortunatelly the manuscript lacks in well structured information, and the phrases are not so well designed.

Response: the manuscript was revised carefully and subjected to english editing

The manuscript is original and well defined and  the results provide an advance in current knowledge. The results are being interpreted appropriately and are significant, but the writting techniques and the structure of the phrases are not well designed, and therefore, it is very difficult to read the article and to understand the meaning of the study and even the meaning of the phrases.  So unfortunatelly, the article is  not written in an appropriate way. 

Response: the manuscript was revised, rephrased carefully and subjected to english editing

The data are robust enough, but it is very difficult to follow the conclusions due to the writting techniques. 

Response: the manuscript subjected to english editing

Surely the paper will attract a wide readership, but only with the corrections being made. The English language is not appropriate and has many writting mistakes, and so, it is imperative to be corrected so that the article could be well designed.

Response: the manuscript subjected to english editing

I have some things to add in the lines below, but unfortunatelly, from my point of view, the article must be rewritten more carefully, especially regarding the English language: 

Line 13: 70, not „seventy”

Response: done as requested

Line 14: 1 ml of ethanol, not „1 ml ethanol”

Response: done as requested

Line 14: „.” after „ethanol”

Response: done as requested

Line 14: without „,” before „and GIII”

Response: done as requested

Line 14: were treated, not „treated”

Response: done as requested

Line 14: 100mg/kg, not „(100mg/kg)

Response: done as requested

Line 14: were treated, not „treated”

Response: done as requested

Line 15: 25mg/kg, not „(25mg/kg)”

Line 15: GVI was given a mixture of a high dose of BPA and NP, not „was given mixture (BPA and NP) of high dose”

Response: done as requested

Line 16: BPA and NP, not „(BPA and NP)”

Response: done as requested

Line 16: A significant elevation in the TNF Alpha, TNF Beta and Caspase 3 serum was recorded individually and also in the groups treated with the mixture of high doses, not „Significant elevation in serum TNF Alpha, TNF Beta and Caspase 3 was recorded individually and in a mixture of high doses treated groups”

Response: done as requested

Line 19: Furthermore, not „Further”

Response: done as requested

Line 19: supports that the exposure, not „support that exposure”

Response: done as requested

Line 24: 70, not „seventy”

Response: done as requested

Line 24: 1 ml of ethanol, not „1 ml ethanol”

Response: done as requested

Line 24: without „,” before „and GIII”

Response: done as requested

Line 25: were treated, not „treated”

Response: done as requested

Line 26: 25mg/kg, not „(25mg/kg)”

Response: done as requested

Line 26: GVI was given a mixture of a high dose of BPA and NP, not „was given mixture (BPA and NP) of high dose”

Response: done as requested

Line 27: BPA and NP, not „(BPA and NP)”

Response: done as requested

Line 29: „,” after „determined”

Response: done as requested

Line 30: The results obtained, not „Obtained results”

Response: done as requested

Line 33: A significant elevation in the TNF Alpha, TNF Beta and Caspase 3 serum was recorded individually and also in the groups treated with the mixture of high doses, not „Significant elevation in serum TNF Alpha, TNF Beta and Caspase 3 was recorded individually and in a mixture of high doses treated groups”

Response: done as requested

Line 34: The disturbance is represented by the histological, not „A disturbance represents the histological”

Response: done as requested

Line 36: The effects, not „Effects”

Response: done as requested

Line 36: „,” after „mixture doses”

Response: done as requested

Line 37: the exposure, not „exposure”

Response: done as requested

Line 38: Furthermore, not „Further”

Response: done as requested

Line 39: supports that the exposure, not „support that exposure”

Response: done as requested

Line 50: were found both in vivo, not „were found in both in vivo”

Response: done as requested

Line 56: involving even baby, not „involving almost baby”

Response: done as requested

Line 57: BPA can be introduced, not „BPA could be introduced”

Response: done as requested

Line 72: to be in contact, not „to contact”

Response: done as requested

Line 81: in females, it can induce early puberty, ovary, breast or endometrium cancer, not „in females, it can induce (early puberty in females, ovary, breast cancer, and endometrium)”

Response: done as requested

Line 82: in males, it can induce prostate, sperm, seminiferous problems, damage in the gonads, in the testis, epididymous lumen and in the Sertoli cells, or cryptorchidism, not „in males, it can induce (prostate, sperm, seminiferous problems, testis, epididymous lumen and epitelial, cryptorchidism, and Sertoli cells) damage in gonads”

Response: done as requested

Line 84: samples, not „sample”

Response: done as requested

Line 88: the effects, not „effects”

Response: done as requested

Line 108: 70, not „seventy”

Response: done as requested

Line 108: without „that rats”

Response: done as requested

Line 109: „King”, not „king”

Response: done as requested

Line 110: of the university, not „university”

Response: done as requested

Line 111: each group containing, not „each group contains”

Response: done as requested

Line 113: were given water, not „were taken water”

Response: done as requested

Line 114: were supplied, not „supplied”

Response: done as requested

Line 115: were supplied, not „supplied”

Response: done as requested

Line 116: were supplied, not „supplied”

Response: done as requested

Line 117: were supplied, not „supplied”

Response: done as requested

Line 119: were supplied, not „supplied”

Response: done as requested

Line 124: of the experiment, not „of experiment”

Response: done as requested

Line 125: retro-orbital, not „retro orbital”

Response: done as requested

Line 126: 2000G, not „2000g”

Response: done as requested

Line 126: the serums were, not „sera were”

Response: done as requested

Line 129: without „()”

Response: done as requested

Line 132: utilized, not „utilize”

Response: done as requested

Line 146: used this Quantitative..., not „this Quantitative...”

Response: done as requested

Line 146: only 1 space between „this” and „Quantitative”

Response: done as requested

Line 151: „,” after „saline”

Response: done as requested

Line 153: followed, not „follow”

Response: done as requested

Line 164: the percentage, not „%”

Response: done as requested

Line 166: had the same, not „has the same”

Response: done as requested

Line 184: „,” after „mixture”

Response: done as requested

Line 184: regarding, not „as regards”

Response: done as requested

Line 191: „,” after „groups”

Response: done as requested

Line 191: significant, not „Significant”

Response: done as requested

Line 201: that the treatment, not „that treatment”

Response: done as requested

Serum Testis Biomarkers

Lines 201-203: The meaning of the paragraph is not clear. Maybe it should have been written: „shows the treatment”, instead of „declare that treatment”

Response: done as requested

Line 203: Regarding the testosterone hormone, not „Testosterone hormone”

Response: done as requested

Line 222: high dose, not „high does’”

Response: done as requested

Line 222: induced, not „induce”

Response: done as requested

Line 236: „,” after „animals”

Response: done as requested

Line 243: were degenerated, not „are degenerated”

Response: done as requested

Line 246: There was an increase, not „An increase”

Response: done as requested

Line 267: and vertical shape, not „and vertical shape”

Response: done as requested

Line 268: shows that, not „show”

Response: done as requested

Line 268: without „,” before „some”

Response: done as requested

Line 331: which induces, not „which induce”

Response: done as requested

Line 337: marker, not „mark”

Response: done as requested

Line 338: „,” before „can destroy”

Response: done as requested

Line 340: have shown, not „was shown”

Response: done as requested

Line 345: without „look”

Response: done as requested

Line 345: activities in every cell, tissue..., not „activities look to be very cell, tissue...”

Response: done as requested

Line 349: in people, not „in peoples”

Response: done as requested

Line 353: elevate, not „elevation”

Response: done as requested

Line 355: „,” after „antioxidant”

Response: done as requested

Line 372: „,” before „while”, instead of „.” before „while”

Response: done as requested

Line 377: „,” after „appeared”

Response: done as requested

Line 398: the endocrin system, not „the endocrin”

Response: done as requested

Line 418: reported that, not „reported”

Response: done as requested

Line 425: and also revealed, not „.” after „[23]”

Response: done as requested

Line 457: by blocking, not „blocking ”

Response: done as requested

Line 463: the action of edocrin, not „the work of edocrin”

Response: done as requested

Line 464: by causing, not „causing”

Response: done as requested

Line 472: that the exposure, not „that exposure”

Response: done as requested

Line 472: „,” after „mice”

Response: done as requested

Line 486: that the exposure, not „that exposure”

Response: done as requested

Line 488: „.” after „apoptosis”

Response: done as requested

Line 488: Also, not „also”

Response: done as requested

Lines 493-494: The exposure of the rats, not „Exposure of rats”

Response: done as requested

Line 501: that the exposure, not „that exposure”

Response: done as requested

Line 519: that the exposure, not „that exposure”

Response: done as requested

Reviewer 2 Report

The manuscript “Disturbance in Some Fertility Biomarkers induced and Changes in Testis Architecture by Chronic Exposure to Various Dosages of each of Nonylphenol or Bisphenol A and their Mix” explores a very interesting topic, considering that Nonylphenol and Bisphenol A are widely used in our everyday life, and it is important to understand how they affect the male fertility. However, there are some issues that need to be revised, as follows:

Comments:

Lines 23-26: explanation of the study groups should be simpler in the abstract.

Line 36: the sentence is quite confusing.

The first paragraph of the introduction section is confusing. The authors refer to plastics and the different types, the endocrine-disrupting chemicals worry, introduce OP, NP and BPA, and then refer “They could be either natural or synthetic chemical materials”. Who are “they”? Are OP, NP and BPA either natural or synthetic? This sentence does not fit there. Also, the authors do not need to describe the abbreviations more than once.

The authors introduce octylphenol but do not analyse its effects in their research. Is there any explanation?

The exposure routes of BPA and the organs/biological samples where it has been found should be added in the introduction section.

Line 115: The authors should correct the “elevated dosage” to “low dosage”.

Line 129: Remove the parentheses in the (Yoshioka et al., 1979) and include the number of the reference.

The abbreviation for each group name is not consistent through the manuscript, for example, in the material section elevated dosage of Bisphenol A is “H BPA” for while in the results section it is “BH”.

The representative letters in figure 2 are not noticeable, the authors should improve this.

All images of figure 2 should be arranged in one panel, so it is easier for the reader to follow and compare all the images.

Lines 239-241: In this sentence, the authors refer to the Bisphenol, Nonylphenol and their mixture in separate or only the mixture Bisphenol with Nonylphenol? It is confusing.

Lines 605, 653, 709: Why are the authors names in references 36 and 81, and the title of the paper in reference 58 in capital letters? The formatting of references must be the same in all of them.

The written English should be revised through all the manuscript. There are innumerable mistakes and grammar errors, including the images.

Author Response

Reviewer 2#

Comments and Suggestions for Authors

The manuscript “Disturbance in Some Fertility Biomarkers induced and Changes in Testis Architecture by Chronic Exposure to Various Dosages of each of Nonylphenol or Bisphenol A and their Mix” explores a very interesting topic, considering that Nonylphenol and Bisphenol A are widely used in our everyday life, and it is important to understand how they affect the male fertility. However, there are some issues that need to be revised, as follows:

Response: thanks for your valuable comments.

Comments:

Lines 23-26: explanation of the study groups should be simpler in the abstract.

Response: done as requested

Line 36: the sentence is quite confusing.

Response: it was rephrased accordingly

The first paragraph of the introduction section is confusing. The authors refer to plastics and the different types, the endocrine-disrupting chemicals worry, introduce OP, NP and BPA, and then refer “They could be either natural or synthetic chemical materials”. Who are “they”? Are OP, NP and BPA either natural or synthetic? This sentence does not fit there. Also, the authors do not need to describe the abbreviations more than once.

Response: They could be either natural or synthetic chemical materials, this sentence was miss typed and was removed from the text accordingly. The repeated abbreviations were removed.

The authors introduce octylphenol but do not analyse its effects in their research. Is there any explanation?

Response:  it was just mentioned as an example and it was removed from the text accordingly

The exposure routes of BPA and the organs/biological samples where it has been found should be added in the introduction section.

Response: “BPA can be introduced to the human body via reusable dental sealants, baby bottles, liquid of canned vegetables, and food packing materials [6].” This paragraph was mentioned in the introduction

Line 115: The authors should correct the “elevated dosage” to “low dosage”.

Response: done as requested

Line 129: Remove the parentheses in the (Yoshioka et al., 1979) and include the number of the reference.

Response: done as requested

The abbreviation for each group name is not consistent through the manuscript, for example, in the material section elevated dosage of Bisphenol A is “H BPA” for while in the results section it is “BH”.

Response: it was uniformed as BH

The representative letters in figure 2 are not noticeable, the authors should improve this.

Response: done as requested

All images of figure 2 should be arranged in one panel, so it is easier for the reader to follow and compare all the images.

Response:  done as requested

Lines 239-241: In this sentence, the authors refer to the Bisphenol, Nonylphenol and their mixture in separate or only the mixture Bisphenol with Nonylphenol? It is confusing.

Response: it was rephrased as “On the other hand, different levels of severity and damage were observed in the testes of the adult rat’s gavage to low doses of 25 mg/kg (BW) of Bisphenol and Nonylphenol in separate and their mixture”

Lines 605, 653, 709: Why are the authors names in references 36 and 81, and the title of the paper in reference 58 in capital letters? The formatting of references must be the same in all of them.

Response: the references number 36, 81 were formatted accordingly

The written English should be revised through all the manuscript. There are innumerable mistakes and grammar errors, including the images.

Response:  done as requested

Round 2

Reviewer 1 Report

 Regarding the structure and accuracy of the phrases, after the corrections were made, the manuscript has indeed well structured information and the phrases are well designed.